# The Cone of Silence: Speech Separation by Localization

**Teerapat Jenrungrot**\* **Vivek Jayaram**\* **Steve Seitz** **Ira Kemelmacher-Shlizerman**
University of Washington
{tjenrung, vjayaram, seitz, kemelmi}@cs.washington.edu

## Abstract

Given a multi-microphone recording of an unknown number of speakers talking concurrently, we simultaneously localize the sources and separate the individual speakers. At the core of our method is a deep network, in the waveform domain, which isolates sources within an angular region $\theta \pm w/2$, given an angle of interest $\theta$ and angular window size $w$. By exponentially decreasing $w$, we can perform a binary search to localize and separate all sources in logarithmic time. Our algorithm allows for an arbitrary number of potentially moving speakers at test time, including more speakers than seen during training. Experiments demonstrate state-of-the-art performance for both source separation and source localization, particularly in high levels of background noise.

## 1 Introduction

The ability of humans to separate and localize sounds in noisy environments is a remarkable phenomenon known as the "cocktail party effect." However, our natural ability only goes so far – we may still have trouble hearing a conversation partner in a noisy restaurant or during a call with other speakers in the background. One can imagine future earbuds or hearing aids that *selectively* cancel audio sources that you don't want to listen to. As a step towards this goal, we introduce a deep neural network technique that can be steered to any direction at run time, cancelling all audio sources outside a specified angular window, aka *cone of silence* (CoS) [1].

But how do you know what direction to listen to? We further show that this directionally sensitive CoS network can be used as a building block to yield simple yet powerful solutions to 1) sound localization, and 2) audio source separation. Our experimental evaluation demonstrates state of the art performance in both domains. Furthermore, our ability to handle an unknown number of potentially moving sound sources combined with fast performance represents additional steps forward in generality. Audio demos can be found at our project website.[2]

We are particularly motivated by the recent increase of multi-microphone devices in everyday settings. This includes headphones, hearing aids, smart home devices, and many laptops. Indeed, most of these devices already employ directional sensitivity both in the design of the individual microphones and in the way they are combined together. In practice however, this directional sensitivity is limited to either being hard tuned to a fixed range of directions (e.g., cardioid), or providing only limited attenuation of audio outside that range (e.g., beam-forming). In contrast, our CoS approach enables true *cancellation* of audio sources outside a specified angular window that can be specified (and instantly changed) in software.

Our approach uses a novel deep network that can separate sources in the waveform domain within any angular region $\theta \pm \frac{w}{2}$, parameterized by a direction of interest $\theta$ and angular window size $w$. For simplicity, we focus only on azimuth angles, but the method could equally be applied to elevation

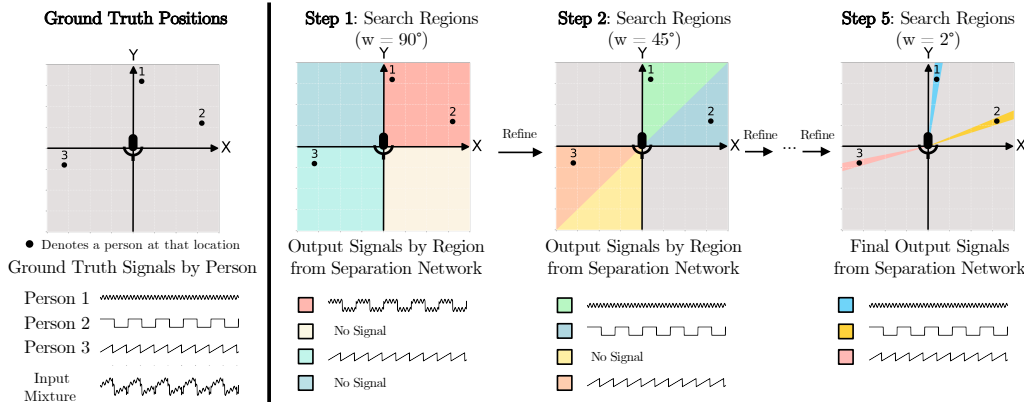

Figure 1: Overview of *Separation by Localization* running binary search on an example scenario with 3 sources. Each panel shows the spatial layout of the scene with the microphone array located at the center. During Step 1, the algorithm performs separation on candidate regions of $90°$. The quadrants with no sound get suppressed and disregarded. The algorithm continues doing separation on smaller partitions of candidate regions until reaching the final step where the angular window size is $2°$.

as well. By exponentially decreasing $w$, we perform a binary search to separate and localize all sources in logarithmic time (Figure 1). Unlike many traditional methods that perform direction based separation, we can also ignore background source types, such as music or ambient noise. Qualitative and quantitative results show state-of-the-art performance and a direct applicability to a wide variety of real world scenarios. Our key contribution is a logarithmic time algorithm for simultaneous localization and separation of speakers, particularly in high levels of noise, allowing for arbitrary number of speakers at test time, including more speakers than seen during training. We strongly encourage the reader to view our supplementary results for a demo of our method and audio results.

## 2 Related Work

Source separation has seen tremendous progress in recent years, particularly with the increasing popularity of learning methods, which improve over traditional methods such as [2, 3]. In particular, *unsupervised source modeling* methods train a model for each source type and apply the model to the mixture for separation by using methods like NMF [4, 5], clustering [6, 7, 8], or bayesian methods [9, 10, 11]. *Supervised source modeling* methods train a model for each source from annotated isolated signals of each source type, e.g., pitch information for music [12]. *Separation based training* methods like [13, 14, 15] employ deep neural networks to learn source separation from mixtures given the ground truth signals as training data, also known as the mix-and-separate framework.

Recent trends include the move to operating directly on waveforms [16, 17, 18] yielding performance improvements over frequency-domain spectrogram techniques such as [19, 20, 21, 6, 22, 23, 24]. A second trend is increasing the numbers of microphones, as methods based on multi-channel microphone arrays [22, 23, 25] and binaural recordings [24, 26] perform better than single-channel source separation techniques. Combined *audio-visual* techniques [27, 28] have also shown promise.

Sound localization is a second active research area, often posed as direction of arrival (DOA) estimation [29, 30, 31]. Popular methods include beamforming [32], subspace methods [33, 34, 35, 36], and sampling-based methods [37]. A recent trend is the use of deep neural networks for multi-source DOA estimation, e.g., [38, 39].

One key challenge is that the number of speakers in real world scenarios is often unknown or non-constant. Many methods require a priori knowledge about the number of sources, e.g., [17, 40]. Recent deep learning methods that address separation with an unknown number of speakers include [41], [42], and [43]. However, these methods use an additional model to help predict the number of speakers, and [43] further uses a different separation model for different numbers of speakers.

Although DOA provides a practical approach for source separation, methods that take this approach suffer from another shortcoming: the direction of interest needs to be known in advance [44]. Without a known DOA for each source, these methods must perform a linear sweep of the entire angular

space, which is computationally infeasible for state-of-the-art deep networks at fine-grained angular resolutions.

Some prior work has addressed joint localization and separation. For example, [45, 46, 47, 48, 49, 50] use expectation maximization to iteratively localize and separate sources. [51] uses the idea of Directional NMF, while [52] poses separation and localization as a Bayesian inference problem based on inter-microphone phase differences. Our method improves on these approaches by combining deep learning in the waveform domain with efficient search.

## 3 Method

In this section we describe our Cone of Silence network for angle based separation. The target angle $\theta$ and window size $w$ are learned independently; Separation at $\theta$ is handled entirely by a pre-shift step, while an additional network input is used to produce the window of size $w$. We also describe how to use the network for *separation by localization* via binary search.

**Problem Formulation**: Given a known-configuration microphone array with $M$ microphones and $M > 1$, the problem of $M$-channel source separation and localization can be formulated in terms of estimating $N$ sources $\mathbf{s}_1, \ldots, \mathbf{s}_N \in \mathbb{R}^{M \times T}$ and their corresponding angular position $\theta_1, \ldots, \theta_N$ from an $M$-channel discrete waveform of the mixture $\mathbf{x} \in \mathbb{R}^{M \times T}$ of length $T$, where

$$\mathbf{x} = \sum_{i=1}^{N} \mathbf{s}_i + \mathbf{bg}. \tag{1}$$

Here $\mathbf{bg}$ represents the background signal, which could be a point source like music or diffuse-field background noise without any specific location.

In this paper we explore circular microphone arrays, but we also describe possible modifications to support linear arrays. The center of our coordinate system is always the center of the microphone array, and the angular position of each source, $\theta_i$, is defined based on this coordinate system. In the problem formulation we assume the sources are stationary, but we describe how to handle potentially moving sources in Section 4.5. In addition, we only focus on separation and localization by azimuth angle, meaning that we assume the sources have roughly the same elevation angle. As we show in the experimental section, this assumption is valid for most real world scenarios.

### 3.1 Cone of Silence Network (CoS)

We propose a network that performs source separation given an angle of interest $\theta$ and an angular window size $w$. The network is tasked with separating speech only coming from azimuthal directions between $\theta - \frac{w}{2}$ and $\theta + \frac{w}{2}$ and disregarding speech coming from other directions. In the following sections we describe how to create a network with this property. Figure 2 shows our proposed network architecture. $\theta$ and $w$ are encoded in a shifted input $\mathbf{x}'$ and a one-hot vector $\mathbf{h}$ as described in Section 3.1.2 and Section 3.1.3 respectively.

#### 3.1.1 Base Architecture

Our CoS network is adapted from the Demucs architecture [53], a music separation network, which is similar to the Wave U-Net architecture [14]. We extend the original Demucs network to our problem formulation by modifying the number of input and output channels to match the number of microphones.

There are several reasons why this base architecture is well suited for our task. As mentioned in Section 2, networks that operate on the raw waveform have been recently shown to outperform spectrogram based methods. In addition, Demucs was specifically designed to work at sampling rates as high as $44.1\,\mathrm{kHz}$, while other speech separation networks operate at rates as low as $8\,\mathrm{kHz}$. Although human speech can be represented at lower sampling rates, we find that operating at higher sampling rates is beneficial for capturing small time difference of arrivals between the microphones. This would also allow our method to be extended to high resolution source types, like music, where a high sampling rate is necessary.

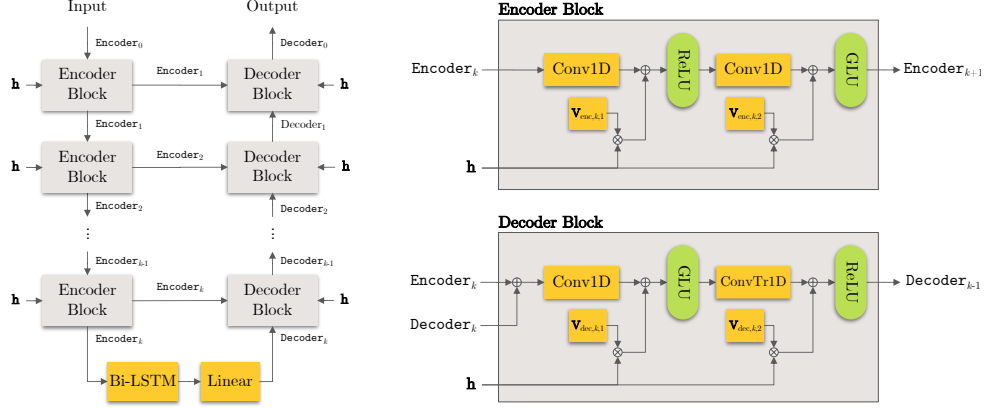

Figure 2: (*left*) our network architecture, (*top-right*) the encoder block, (*bottom-right*) the decoder block. In all diagrams, $\mathbf{h}$ refers to the global conditioning variable corresponding to an angular window size $w$.

### 3.1.2 Target Angle $\theta$

In order to make the network output sources from a specific target angle $\theta$, we use a shifted mixture $\mathbf{x}' \in \mathbb{R}^{M \times T}$ based on $\theta$. We found that that this worked better than trying to directly condition the network based on both $\theta$ and $w$. $\mathbf{x}'$ is created as follows: by calculating the time difference of arrival at each microphone, we shift each channel in the original signal $\mathbf{x}$ such that signals coming from angle $\theta$ are temporally aligned across all $M$ channels in $\mathbf{x}'$.

We use the fact that the time differences of arrival (TDOA) between the microphones are primarily based on the azimuthal angle for a far-field source. This assumption is valid when the sources are roughly on the same plane as the microphone array [54]. Let $c$ be the speed of sound, $sr$ be our sampling rate, $p_\theta$ be the position of a far-field source at angle $\theta$, and $d(\cdot, \cdot)$ be a simple Euclidean distance. The TDOA in samples for the source to reach the $i$-th microphone is

$$T_{\texttt{delay}}(p_\theta, \texttt{mic}_i) = \left\lfloor \frac{d(p_\theta, \texttt{mic}_i)}{c} \cdot sr \right\rfloor \tag{2}$$

In our experiments, we chose $\texttt{mic}_0$ as our canonical position, meaning that $\mathbf{x}'_0 = \mathbf{x}_0$ and all other channels $\mathbf{x}'_i$ are shifted to align with $\mathbf{x}'_0$.

$$\mathbf{x}'_i = \texttt{shift}(\mathbf{x}_i, T_{\texttt{delay}}(p_\theta, \texttt{mic}_0) - T_{\texttt{delay}}(p_\theta, \texttt{mic}_i)) \quad i = 1, \ldots, M-1 \tag{3}$$

$\texttt{shift}$ is a 1-D shift operation with one sided zero padding. This idea is similar to the first step of a Delay and Sum Beamformer [55]. We then train the network to output sources which are temporally aligned in $\mathbf{x}'$ while ignoring all others. For $M > 2$, these shifts are unique for a specific angle $\theta$, so sources from other angles will not be temporally aligned. If $M = 2$ or the mic array is linear, then sources at angles $\theta$ and $-\theta$ have the same per-channel shift leading to front-back confusion.

### 3.1.3 Angular Window Size $w$

Although the network trained on shifted inputs as described in Section 3.1.2 can produce accurate output for a given angle $\theta$, it requires prior knowledge about the target angle $\theta$ for each source of interest. In addition, real sources are not perfect point sources and have finite width, especially in the presence of slight movements.

To solve these problems, we introduce a second variable, an angular window size $w$ which is passed as a global conditioning parameter to the network. This angular window size facilitates the application of this network for a fast binary search approach. It also allows the localization and separation of moving sources. By using a larger angular window size and a smaller temporal input waveform, it is possible to localize and separate moving sources within that window.

Motivated by the global conditioning framework in WaveNet [56], we use a one-hot encoded variable $\mathbf{h}$ to represent different window sizes. In our experiments, we use $\mathbf{h}$ of size 5 corresponding to

window sizes from the set $\{90°, 45°, 23°, 12°, 2°\}$. By passing $\mathbf{h}$ to the network with our shifted input $\mathbf{x}'$, we can explicitly make the network separate sources from the region $\theta \pm \frac{w}{2}$. We embed $\mathbf{h}$ to all encoder and decoder blocks in the network, using a learning linear projection $\mathbf{V}_{\cdot,k,\cdot}$ as shown in Figure 2. Formally, the equations for the encoder block and decoder block can be written as follows:

$$\begin{aligned} \texttt{Encoder}_{k+1} = \text{GLU}(\mathbf{W}_{\text{encoder},k,2} * \text{ReLU}(\mathbf{W}_{\text{encoder},k,1} * \texttt{Encoder}_k \\ + \mathbf{V}_{\text{encoder},k,1}\mathbf{h}) + \mathbf{V}_{\text{encoder},k,2}\mathbf{h}), \end{aligned} \quad (4)$$

$$\begin{aligned} \texttt{Decoder}_{k-1} = \text{ReLU}(\mathbf{W}_{\text{decoder},k,2} *^\top \text{GLU}(\mathbf{W}_{\text{decoder},k,1} * (\texttt{Encoder}_k + \texttt{Decoder}_k) \\ + \mathbf{V}_{\text{decoder},k,1}\mathbf{h}) + \mathbf{V}_{\text{decoder},k,2}\mathbf{h}). \end{aligned} \quad (5)$$

The notation $\mathbf{W}_{\cdot,k,\cdot} * \mathbf{x}$ denotes a 1-D convolution between the weights for the layer of an encoding/decoding block at level $k$ and an input $\mathbf{x}$. The notation $*^\top$ denotes a transposed convolution operation. Empirically we found that passing $\mathbf{h}$ to every encoder and decoder block worked significantly better than passing it to the network only once. Evidence that the CoS network learns the desired window size is presented in Figure 4.

### 3.1.4 Network Training

Consider an input mixture $\mathbf{x}$ of $N$ sources $\mathbf{s}_1, \ldots, \mathbf{s}_N$ with the corresponding locations $\theta_1, \ldots, \theta_N$ along with a target angle $\theta_t$ and window size $w$. The network is trained with the following objective function:

$$\mathcal{L}(\mathbf{x}; \mathbf{s}_1, \ldots, \mathbf{s}_N, \theta_t, w) = \left\| \tilde{\mathbf{x}}' - \sum_{i=1}^{N} \mathbf{s}_i' \cdot \mathbb{I}\left(\theta_t - \frac{w}{2} \leq \theta_i < \theta_t + \frac{w}{2}\right) \right\|_1 \quad (6)$$

where $\mathbf{x}'$ and $\mathbf{s}_i'$ are the shifted signals of the input mixture and ground truth signal as described in Section 3.1.2 based on the target angle $\theta_t$. $\tilde{\mathbf{x}}'$ is the output of the network using the shifted signal $\mathbf{x}'$ and the angular window $w$. $\mathbb{I}(\cdot)$ is an indicator function, indicating whether $\mathbf{s}_i$ is present in the region $\theta_t \pm \frac{w}{2}$. If no source is present in the region $\theta_t \pm \frac{w}{2}$, the training target is a zero tensor $\mathbf{0}$.

## 3.2 Localization and Separation via Binary Search

By starting with a large window size $w$ and decreasing it exponentially, we can perform a binary search of the angular space in logarithmic time, while separating the sources simultaneously. More concretely, we start with our initial window size $w_0 = 90°$, our initial target angles $\theta_0 = \{-135°, -45°, 45°, 135°\}$, and our observed $M$-channel mixture $\mathbf{x} \in \mathbb{R}^{M \times T}$. In the first pass we run the network $\text{CoS}(\mathbf{x}', w_0)$ for all $\theta_0^i \in \theta_0$. This first step is the quadrant based separation illustrated in Step 1 of Figure 1. Because regions without sources will produce empty outputs, we can discard large angular regions early on with a simple cutoff. We then regress on a smaller window size, $w_1 = 45°$ and the new candidate regions $\theta_1 = \bigcup_i \{\theta_0^i \pm \lfloor \frac{w_0}{2} \rfloor\}$ for $\theta_0^i$ regions with high energy outputs from $\text{CoS}(\mathbf{x}', w_0)$. We continue to regress on smaller window sizes until reaching the desired resolution. The complete algorithm is written below and shown in Figure 1.

---

**Algorithm 1:** Separation by Localization via Binary Search

---

**Inputs** : $M$-channel input mixture $\mathbf{x} \in \mathbb{R}^{M \times T}$ and the microphone array position $\{\texttt{mic}_i\}_{i=0}^{M-1}$
**Output** : Separated signals and their locations

$\textsc{separateAndLocalize}\,(\mathbf{x}, \{\texttt{mic}_i\}_{i=0}^{M-1})$

1   Initialize $L$, $w_{0,\ldots,L-1}$, and $\theta_0$.
2   **for** $\ell \in \{0, 1, \ldots, L-1\}$ **do**
3     $\theta_{\ell+1} \leftarrow \{\}$
4     **for** $\theta_\ell^i \in \theta_\ell$ **do**
5       $\mathbf{x}' \leftarrow \textsc{preShift}(\mathbf{x}, \theta_\ell^i, \{\texttt{mic}_j\}_{j=0}^{M-1})$
6       $\tilde{\mathbf{x}}' \leftarrow \text{CoS}(\mathbf{x}', w_\ell)$
7       Update $\theta_{\ell+1}$ accordingly by adding $\theta_\ell^i \pm \lfloor \frac{w_\ell}{2} \rfloor$ to $\theta_{\ell+1}$ if $\tilde{\mathbf{x}}'$ isn't empty.
8     **end**
9   **end**
10   **return** *Non-max suppression on sources at $\theta \in \theta_L$.*

---

To avoid duplicate outputs from adjacent regions, we employ a non-maximum suppression step before outputting the final sources and locations. For this step, we consider both the angular proximity and similarity between the sources. If two outputted sources are physically close and have similar source content, we remove the one with the lower source energy. For example, for outputs $(\tilde{\mathbf{x}}'_i, \theta_i)$ and $(\tilde{\mathbf{x}}'_j, \theta_j)$ with $\|\tilde{\mathbf{x}}'_i\| > \|\tilde{\mathbf{x}}'_j\|$, we remove $(\tilde{\mathbf{x}}'_j, \theta_j)$ if $|\theta_i - \theta_j| < \epsilon_\theta$ and $\|\tilde{\mathbf{x}}'_i - \tilde{\mathbf{x}}'_j\| < \epsilon_x$.

### 3.3 Runtime Analysis

Suppose we have $N$ speakers and the angular space is discretized into $r = \frac{360°}{w}$ angular bins. The binary search algorithm runs for at most $\mathcal{O}(\log r)$ steps and requires at most $\tilde{\mathcal{O}}(N)$ forward passes on every step. Thus, the total number of forward passes is $\mathcal{O}(N \log r)$ while a linear sweep always runs in $\mathcal{O}(r)$ forward passes.

In most cases, $N \ll r$, so the binary search is clearly superior. For instance, when operating at a $2°$ resolution, the average number of forward passes our algorithm takes to separate 2 voices in the presence of background is 32.64, compared to 180 for a linear sweep. A forward pass of the network on a single GPU takes $0.03\,\mathrm{s}$ for a $3\,\mathrm{s}$ input waveform at $44.1\,\mathrm{kHz}$, meaning that the binary search algorithm in this scenario could keep up with real-time while the linear search could not.

## 4  Experiments

In this section, we explain our synthetic dataset and manually collected real dataset. We show numerical results for separation and localization on the synthetic dataset and describe qualitative results on the real dataset.

### 4.1  Synthetic Dataset

Numerical results are demonstrated on synthetically rendered data. To generate the synthetic dataset, we create multi-speaker recordings in simulated environments with reverb and background noises. All voices come from the VCTK dataset [57], and the background samples consist of recordings from either noisy restaurant environments or loud music. The train and test splits are completely independent and there are no overlapping identities or samples. We chose VCTK over other widely used datasets like LibriSpeech [58] and WSJ0 [59] because VCTK is available at a high sampling rate of $48\,\mathrm{kHz}$ compared to $16\,\mathrm{kHz}$ as offered by others. In the supplementary materials, we show results and comparisons with lower sampling rates.

To synthesize a single example, we create a 3-second mixture at $44.1\,\mathrm{kHz}$ by randomly selecting $N$ speech samples and a background segment and placing them at arbitrary locations in a virtual room of a randomly chosen size. We then simulate room impulse responses (RIRs) using the image source method [60] implemented in the `pyroomacoustics` library [61]. To approximate a diffuse-field background noise, the background source is placed further away, and the RIR for the background is generated with high-order images, causing indirect reflections off room walls [62]. All signals are convolved with the corresponding RIRs and rendered to a 6-channel circular microphone array ($M = 6$) of radius $2.85\,\mathrm{in}$ ($7.25\,\mathrm{cm}$). The volumes of the sources are chosen randomly in order to create challenging scenarios; the input SDR is between $-16\,\mathrm{dB}$ and $0\,\mathrm{dB}$ for most of the dataset. For training our network, we use 10,000 examples with $N$ chosen uniformly between 1 and 4, inclusively, at random, and for evaluating we use 1,000 examples with $N$ dependent on the evaluation task.

### 4.2  Source Separation

To evaluate the source separation performance of our method, we create mixtures consisting of 2 voices ($N = 2$) and 1 background, allowing comparisons with deep learning methods that require a fixed number of foreground sources. We use the popular metric *scale-invariant signal-to-distortion ratio* (SI-SDR) [63]. When reporting the increase from the input to output SI-SDR, we use the label SI-SDR improvement (SI-SDRi). For deep learning baselines in the waveform domain we chose TAC [40], a recently proposed neural beamformer, and a multi-channel extension of Conv-TasNet [18], a popular speech separation network. For this multi-channel Conv-TasNet, we changed the number of input channels to match the number of microphones in order to process the full mixture. To compare with spectrogram based methods, we use oracle baselines based on the time-frequency representation

Table 1: Separation Performance. Larger SI-SDRi is better. The SI-SDRi is computed by finding the median of SI-SDR increases from Figure 3.

| Method | SI-SDRi (dB) |
|---|---|
| *Waveform-based* | |
| Conv-TasNet [18] | 15.526 |
| TAC [40] | 15.121 |
| **Ours - Binary Search** | **17.059** |
| Ours - Oracle Location | 17.636 |
| *Spectrogram-based* | |
| Oracle IBM [64, 65] | 13.359 |
| Oracle IRM [64, 66] | 4.193 |
| Oracle MWF [64, 67] | 8.405 |

like Ideal Binary Mask (IBM), Ideal Ratio Mask (IRM), and Multi-channel Wiener Filter (MWF). For more details on oracle baselines, please refer to [64]. Table 1 and Figure 3 show the comparison between our proposed system and the baseline systems.

Notice that our method strongly outperforms the best possible results obtainable with spectrogram masking, and is slightly better than recent deep-learning baselines operating on the waveform domain. Furthermore, our network can accept explicitly known source locations (given by *Ours-Oracle Location*), allowing the separation performance to improve further when the source positions are given.

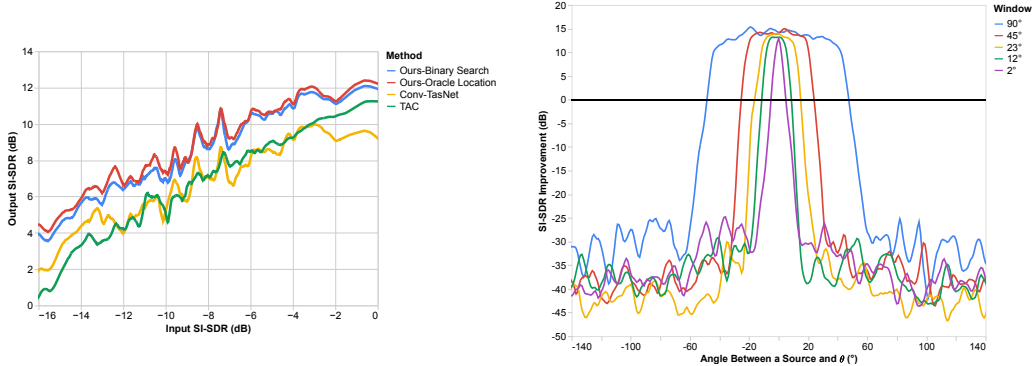

Figure 3: (left) Input SI-SDR vs Output SI-SDR for waveform based methods. Some methods are not shown to improve the visibility.

Figure 4: (right) Evidence that the network amplifies voices between $\theta \pm \frac{w}{2}$ and suppresses all others.

## 4.3 Source Localization

To evaluate the localization performance of our method, we explore two variants of the same dataset in Section 4.2. The first set contains 2 voices and 1 background, exactly as in the previous section, and the second contains 2 voices with no background, a slightly easier variation. Here, we report the CDF curve of the angular error, i.e., the fraction of the test set below a given angle error.

For baselines, we choose popular methods for direction of arrival estimation, both learning-based systems [38] and learning-free systems [33, 32, 34, 35, 37, 36]. For the scenario with 2 voice sources and 1 background source, we let the learning-free baseline algorithms localize 3 sources and choose the 2 sources closest to the ground truth voice locations. This is a less strict evaluation that does not require the algorithm to distinguish between a voice and background source. For the learning-based method [38], we retrained the network separately for each dataset in order to predict the 2 voice locations, even in the presence of a background source. Figure 5 shows the CDF plots for both scenarios.

Table 2: Localization Performance

| Method | Median Angular Error | |
| --- | --- | --- |
| | 2 Voices | 2 Voices + BG |
| *Learning-free* | | |
| MUSIC [33] | 82.5° | 36.8° |
| SRP-PHAT [32] | 6.2° | 46.4° |
| CSSM [34] | 30.1° | 36.3° |
| WAVES [35] | 16.4° | 32.1° |
| FRIDA [37] | 6.9° | 18.5° |
| TOPS [36] | 2.4° | 11.5° |
| *Learning-based* | | |
| MLP-GCC [38] | 1.0° | 41.5° |
| **Ours** | **2.1°** | **3.7°** |

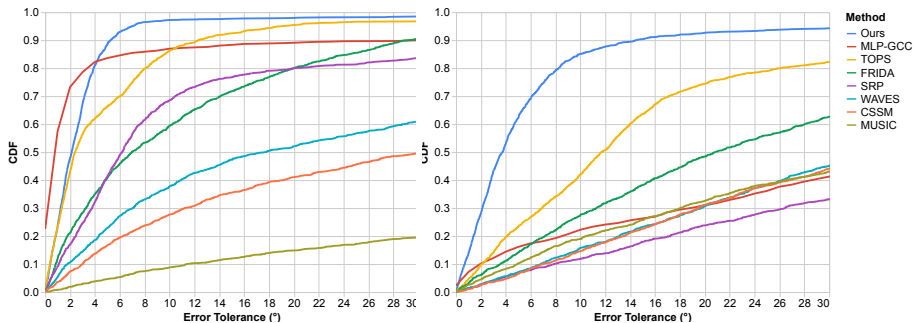

Figure 5: Localization Performance: (*Left*) error tolerance curve on mixtures of 2 voices, (*right*) error tolerance curve on mixtures with 2 voices and 1 background.

Our method shows state-of-the-art performance in the simple scenario with 2 voices, but some baselines show similar performance to ours. However, when background noise is introduced, the gap between our method and the baselines increases greatly. Traditional methods struggle, even when evaluated less strictly than ours, and MLP-GCC [38] cannot effectively learn to distinguish a voice location from background noise.

### 4.4 Varying Number of Speakers

To show that our method generalizes to an arbitrary number of speakers, we evaluate separation and localization on mixtures containing up to 8 speakers with no background. We train the network with mixtures of 1 background and up to 4 voices and evaluate the separation results with median SI-SDRi and the localization performance with median angular error. For a given number of speakers $N$, we take the top $N$ outputs from the network and find the closest permutation between the outputs and ground truth. We report the results in Table 3. Notice that we are reporting results on scenarios where there are more speakers than seen during training.

We report the SI-SDRi and median angular error, together with the precision and recall of localizing the voices within $15°$ of the ground truth when the algorithm has no information about the number of speakers. We remark that as the number of speakers increases, the recall drops as expected. The precision increases are due to the fact that there are fewer false positives when there are many speakers in the scene. The results suggest that our method generalizes and works even in scenarios with more speakers than seen in training.

### 4.5 Results on Real Data and Moving Sources

**Dataset**: To show results on real world examples, we use the ReSpeaker Mic Array v2.0 [68], which contains $M = 4$ microphones in a circle of radius $1.27$ in ($32.2$ mm). Although a network trained

Table 3: Generalization to arbitrary many speakers. We report the separation and localization performance as the number of speakers varies.

| Number of Speakers $N$ | 2 | 3 | 4 | 5 | 6 | 7 | 8 |
|---|---|---|---|---|---|---|---|
| SI-SDRi (dB) | 13.9 | 13.2 | 12.2 | 10.8 | 9.1 | 7.2 | 6.3 |
| Median Angular Error | 2.0° | 2.3° | 2.7° | 3.5° | 4.4° | 5.2° | 6.3° |
| Precision | 0.947 | 0.936 | 0.897 | 0.912 | 0.932 | 0.936 | 0.966 |
| Recall | 0.979 | 0.972 | 0.915 | 0.898 | 0.859 | 0.825 | 0.785 |

purely on synthetic data works well, we find that it is useful to fine-tune with data captured by the microphone. To do this we recorded VCTK samples played over a speaker from known locations, and also recorded a variety of background sounds played over a speaker. We then created mixtures of this real recorded data and jointly re-trained with real and fully synthetic data. Complete details of this capture process are described in the supplementary materials.

**Results**: In the supplementary videos[3] we explore a variety of real world scenarios. These include multiple people talking concurrently and multiple people talking while moving. For example, we show that we can separate people on different phone calls or 2 speakers walking around a table. To separate moving sources, we stop the algorithm at a coarser window size (23°) and use inputs corresponding to 1.5 seconds of audio. With these parameters, we find that it is possible to handle substantial movement because the angular window size captures each source for the duration of the input. We then concatenate sources that are in adjacent regions from one time step to the next. Because our real captured data does not have precise ground truth positions or perfectly clean source signals, numerical results are not as reliable as the synthetic experiment. However, we have included some numerical results on real data in the supplementary materials.

## 4.6 Limitations

There are several limitations of our method. One limitation is that we must reduce the angular resolution to support moving sources. This in contrast to specific speaker tracking methods that can localize moving sources to a greater precision [69, 70]. Another limitation is that in the minimal two-microphone case, our approach is susceptible to front-back confusion. This is an ambiguity that can be resolved with binaural methods that leverage information like HRTFs [71, 72] A final limitation is that we assume the microphone array is rotationally symmetric. For ad-hoc microphone arrays, our pre-shift method would still allow for separation from a known position. However, the angular window size $w$ would have to be learned dependent on $\theta$, making the binary search approach more difficult.

## 5 Conclusion

In this work, we introduced a novel method for joint localization and separation of audio sources in the waveform domain. Experimental results showed state-of-the-art results, and the ability to generalize to an arbitrary number of speakers, including more than seen during training. We described how to create a network that separates sources within a specific angular region, and how to use that network for a binary search approach to separation and localization. Examples on real world data also show that our proposed method is applicable to real-life scenarios. Our work also has the potential to be extended beyond speech to perform separation and localization of arbitrary sound types.

## Acknowledgements

The authors thank the labmates from UW GRAIL lab. This work was supported by the UW Reality Lab, Facebook, Google, Futurewei, and Amazon.

## Broader Impact Statement

We believe that our method has the potential to help people hear better in a variety of everyday scenarios. This work could be integrated with headphones, hearing aids, smart home devices, or laptops, to facilitate source separation and localization. Our localization output also provides a more privacy-friendly alternative to camera based detection for applications like robotics or optical tracking. We note that improved ability to separate speakers in noisy environments comes with potential privacy concerns. For example, this method could be used to better hear a conversation at a nearby table in a restaurant. Tracking speakers with microphone input also presents a similar range of privacy concerns as camera based tracking and recognition in everyday environments.

## Footnotes

[2] https://grail.cs.washington.edu/projects/cone-of-silence/

[3]Also available at https://grail.cs.washington.edu/projects/cone-of-silence/

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
