[Supplementary Material · supplementary.pdf]

# Supplementary Materials

Figure 6: Overview of *Separation by Localization*. Similar to the overview figure in the main paper. This figure is color-blind friendly, black and white printing friendly, and photocopy friendly.

## A    Hyperparameters and Training Details

**Rendering Parameters** For the simulated scenes, the origin of the scene is centered on the microphone array. The foregrounds are placed randomly between 1 and 5 meters away from the microphone array while the background is placed between 10 and 20 meters away. The walls of a virtual rectangular room are chosen between 15 and 20 meters away, expanding as necessary so the background is also within the room. The reverb absorption rate of the foreground is randomly chosen between 0.1 and 0.99, while the absorption rate of the background is chosen between 0.5 and 0.99.

**Training Parameters** We use a learning rate of $3 \times 10^{-4}$ and initialized our training from the pretrained single-channel Demucs weights. We use ADAM optimizer [73] for training the network with the following parameters: $\beta_1 = 0.9$, $\beta_2 = 0.999$, and $\epsilon = 10^{-8}$. We found that training on our spatial dataset converged after roughly 20 epochs.

**Data Augmentation** As an additional data augmentation step we make the following perturbations to the data: Gaussian noise is added with a standard deviation of 0.001, and high-shelf and low-shelf gain of up to $2\,\mathrm{dB}$ are randomly added using the sox library[4].

## B    Real Dataset

**Data Collection** In order to fine-tune the network on real data, we played samples over a speaker and recorded the signals with the real microphone array. Approximately 3 hours of VCTK samples were played from a QSC K8 loudspeaker in a quiet room with the speaker volume set approximately to the volume of a human voice. The loudspeaker was placed at carefully measured positions between 1-4 meters away from the microphone array. We used azimuth angles in $30°$ increments for a total of 12 different positions. The elevation angle was roughly the same as the microphone array. We maintained the train and test splits of the VCTK dataset to avoid overlapping identities. Because we could not record true diffuse background noise, we played various background noises over the loudspeaker such as music or recorded restaurant sounds. With these recorded samples, we could create mixtures with access to the ground truth voice samples. We found that jointly training with $50\%$ real and $50\%$ synthetic mixtures gave the best performance.

**Numerical Results** Because we did not have access to a true acoustic chamber, the ground truth samples and positions are not as reliable for evaluation as the fully synthetic data. However, we report separation results on mixtures of 2 voices and 1 background from the test set of real recorded data in Table 4. This, along with the qualitative samples, shows evidence that our method can generalize to real environments. We note that oracle baselines outperform our methods and other waveform-based

Table 4: Separation performance on the real dataset

| Method | Median SI-SDRi (dB) |
|--------|---------------------|
| Ours | 8.885 |
| TAC [40] | 8.427 |
| Conv-TasNet [18] | 6.497 |
| Oracle IBM | 9.220 |
| Oracle IRM | 10.327 |
| Oracle MWF | 9.925 |

baselines because oracle baselines have access to the ground-truth utterances. Additionally, our method outperforms other non-oracle baselines.

## C   Sample waveforms and spectrograms

In this section, we show sample waveforms of an input mixture and separated voices using our method. The input mixture contains two voices and one background, and we show an example of separation results in two different domains: waveform (Figure ??) and time-frequency spectrogram (Figure 8). Although the output closely matches the ground truth, we can see several differences. As illustrated by Figure 8, we observe that the network struggles in regions where the voice's energy is low. Additionally, we find that the network can create artifacts in the high-frequency regions, which is why a simple denoising step or low pass filter is often helpful.

More example audio files are provided in the zip files.

Figure 7: We show an example of separation on an input mixture containing 2 voices and background. The topmost signal is the input mixture. (*top*) input mixture, (*center + bottom*) separated voices.

## D   Sampling Rate

We show the effect of lowering the sample rate on both separation and localization in Table 5. We remark that our separation quality is worse at lower sample rates, showing that our model takes advantage of the higher sample rate.

Figure 8: An example of separation results with 2 voices and 1 background. (top) the spectrogram of an input mixture, (left) the spectrograms of outputs from the network (right) the spectrograms of the ground truth reference voice signals.

Table 5: Separation and localization performances on datasets with different sampling rates

| Method | Sampling Rate | |
|---|---|---|
| | 44.1kHz | 16kHz |
| **Separation: Median SI-SDRi** (dB) | | |
| Ours - Binary Search | 17.059 | 14.132 |
| Ours - Oracle Position | 17.636 | 14.468 |
| TAC [40] | 15.104 | 13.613 |
| Conv-TasNet [18] | 15.526 | 15.559 |
| Oracle IBM | 13.359 | 13.611 |
| Oracle IRM | 4.193 | 4.289 |
| Oracle MWF | 8.405 | 8.893 |
| **Localization: Median Angular Error** (°) | | |
| Ours - 2 Voices 1 BG | 3.73° | 3.98° |
| Ours - 2 Voices No BG | 2.13° | 2.68° |

## Footnotes

[4] http://sox.sourceforge.net/