[Reviews · NeurIPS 2020]

Review 1

Summary and Contributions: This paper introduces a new method for deep learning-based sound source separation, localization and counting in the waveform domain in the presence of background noise and reverberation. The method introduces a number of new ideas, such as encoding angular information via time-shifts at the networks input, conditioning with an angular window size and using tree search to reduce computational time. An extensive set of experiments and comparisons demonstrates the superiority of the proposed method against a number of state-of-the-art learning-based and non-learning based sound separation and localization methods on both simulated and real data.

Strengths: The paper is clearly written and easy to follow. The proposed method brings in new ideas that coud be useful in multichannel audio signal processing beyond this application, like encoding angular information via time-shifts at the network's input, which has the potential of making the method robust to changes in the array geometry (although this is not specifically tested in the paper). The experiments are carefully designed and comparisons with a number of methods representative of the state of the art or oracle-based compellingly demonstrate the superiority of the approach. The provided results on real data are particularly impressive.

Weaknesses: While the paper is generally clearly written, sound, and the work most likely reproducible, there are some missing details and inaccuracies in the technical and experimental sections that need to be addressed, as detailed in the feedback section below.

Correctness: The claims and method are correct (up to a minor inaccuracy mentioned in the feedback section below). The empirical methodology is carefully designed and compellingly demonstrate the strength of the method.

Clarity: The paper is very well written and easy to follow.

Relation to Prior Work: The bibliography is particularly well-furnished and faithfully represents major contributions in the vast fields of sound separation and localization over the past years.

Reproducibility: Yes

Additional Feedback: # After reading the other reviews and authors' rebuttal, I stand by my score of 9. All the reviewers eventually agreed that the work was novel and significant. Please address the comments below: 1) Please use the standard term "time difference of arrival (TDOA)" rather than time of arrival difference. 2) L116: "We used the fact that the time-of-arrival differences between the microphones are solely based on the azimuthal angle for a far field source". This is only true for linear arrays, but not for circular arrays, as used in the paper. For circular arrays, even in the far field, the TDOAs depends on both azimuth and elevation angles (to see this, notice that if a source is right above the array, all the TDOAs are null). If all sources have roughly the same elevation, this claim is approximately true, which may explain why the proposed method still works reasonnably well. An extension of the proposed method to 2D localization seems like a interesting research avenue. 3) Eq. (6): while the authors clearly explain how x' is computed, it is not entirely clear how the s_i' are computed: is it a shift by +Tdelay(p_theta,mic_0)? Please clarify. 4) Some important details are missing in the experimental data description in Section 4.1, as listed below. Some of those could be provided as appendix/supplementary material. - What range of simulated room sizes, surface absoroptions and RT60 were used? - How was the microphone array positioned inside simulated rooms? - How were the sources positioned with respect to the array, and crucially what heights and what distances? - The real data used for fine-tuning are not sufficiently described (number of array-source positions, type of emitted signals, etc..). What was the ratio of real to synthetic data during fine tuning? Was there an overlap between traning and fine-tuning synthetic data? - Not enough details are given on the "spectrogram denoising method [65] [used] as a post-processing step". In particular, results without using post-processing must be reported. - The array geometries used for synthetic data and for real data are different, which is interesting. Is the method expected to generalize to any array geometry? 5) The authors refer to a "multi-channel extension of Conv-TasNet" but do not give any references or explanation for this, please detail how this was done. 6) L271: "For ad-hoc microphone arrays, w would have to be learned dependent on theta" ... and phi (the elevation). Extension to ad-hoc arrays would require an extension to 2D localization, which should be feasible but not trivial. Typos: L22: state of the art performance -> state-of-the-art performance L178: ... our algorithm takes separate 2 voices -> our algorithm takes to separate 2 voices L198: ...in pyroomacoustics library -> in the pyroomacoustics library Fig. 5: please include citations in the figure's legend.


Review 2

Summary and Contributions: This paper describes how to modify a speech separation network to make use of spatial information in order to extract sounds coming from different locations. In order to do so the authors make use of a binary search approach that, when combined with the ability to define beam width, can hone on a source very fast.

Strengths: I really liked this paper. The idea is interesting, the execution is very good, and the results are highly competitive. There is a nice melding of ideas from array signal processing and deep learning, that produces a very interesting contribution.

Weaknesses: The authors seem to be mostly pushing the binary search as a primary contribution. This is not that new a concept. Using binary search with tunable-width beamformers is something that is well-known and commonly used. One could argue that swapping the beamformer with a neural network isn't that novel (although, knowing that this can work that well is indeed a very welcome contribution). It also isn't entirely clear what is the practical implication of this work (since this isn't a theory paper). When a new source enters a scene, one can use this binary search approach to quickly hone on it. However, in most realistic scenarios new sources do not come too often, and once a beamformer locks onto a source there is no need for additional global scanning (we can follow up on position changes with tracking). For a highly chaotic situation one might need to perform broad scans like that more often, but it isn't clear how much of a benefit this approach offers. This isn't a big deal, but I would have liked a short discussion of this. Also, the use of a BLSTM prohibits a real-time implementation since it is not causal, so talking about efficiency for a system that won't be under time pressure seems a little moot.

Correctness: I did not find any problems on this front. The paper seems fine.

Clarity: The writing was very clear, and easy to follow. Kudos to the authors for doing so.

Relation to Prior Work: There is a very extensive review of related methods and the authors provide relevant comparisons. One thing that I would have liked to see is a comparison with a state-of-the-art beamformer. Being able to show how much better such systems are when compared to their shallow and linear predecessors is important in order to put them into context.

Reproducibility: Yes

Additional Feedback: The rebuttal did not change my evaluation, I still think this is a good paper.


Review 3

Summary and Contributions: This paper proposes to simultaneously localize sound sources and separate individual speakers. The authors operate in the wave domain by searching and shrinking the areas of interest in a curriculum manner. The idea is novel and interesting. The method shows strong performances given recorded audios at high sampling rates.

Strengths: ++ This paper is well-written. ++ The idea of this paper is interesting, the method is quite straight forward but the solution looks nice. ++ The results can outperform strong competing methods on the specific dataset. ++ The non-maximum suppression used is reasonable for finetuning the result. ++ Code is provided.

Weaknesses: Although I am fond of the method proposed, there are certain issues with the experiment settings: -- This paper uses high sampling rate at 44.1kHz, while the normal setting for most works are 16kHz. Such high sampling rate is also not popular in real-world scenarios (As the authors cannot even use the WSJ dataset). It is easy to understand that “higher sampling rates is beneficial for capturing small time of arrival differences”. The reviewer would like to know the performance drop and comparison with other methods when the sampling rate is 16kHz. -- The target audio is also required to be clean according to the details. “our training requires a clean reference utterance of each target speaker.” This is actually a strong restriction to training samples. Why cannot the samples be mixed with reverberations? -- Given the above reasons and the fact that for real-world testing, the audios are also selected from VCTK. It is highly possible that the network is overfitted to the dataset. What is the composition of the train/val/test settings? Are there any overlaps over identities? Could the authors report testing results with audios selected from another dataset? Or real-world recordings? -- The result of Oracle IRM is amazingly low. Is it correct?

Correctness: Yes.

Clarity: Yes.

Relation to Prior Work: Yes.

Reproducibility: Yes

Additional Feedback: Overall, I am in favor of the paper, however, I would like to see the authors' explanations to the problems listed in the weakness part. More comments that do not affect the reviewer's judgment: 1. The spectrogram denoising method is used as a post-processing step in real-data demos. Is this post-processing used in synthetic testing? 2. It would be better if the authors could provide the numbers of pesq and stoi. 3. There are some typos: L112 “We found that that this worked better than trying to directly condition the network based on both θ and w. x 0 is created as follows.” ------------------------------------------------------------------------------------------------------ I have read the authors' rebuttal. Under the condition that the authors are willing to add results of lower sampling rate experiments to the final version, I would like to support the acceptance of this paper. Please also add the promised discussions to other reviewers.


Review 4

Summary and Contributions: Given a microphone array data, the authors design a network which can map the multi-channel audio from an array to the audio of a source at desired angular region. They then propose to use this network to sweep the azimuthal space to localize the source. Essentially a delay-and-sum beamformed signal corresponding to a specific location is input to the network and the network is trained to produce the output waveform corresponding to a source at that location. The authors then do a logarithmic sweep at different resolutions to localize a source as well as obtain a separated output corresponding to each source.

Strengths: This idea of beam-forming and sweeping to get source location and source separation is quite common in signal processing but what is novel is the authors do this using a deep network.

Weaknesses: 1. The authors start with an unrealistic and rather simple signal model. They do not even account for signal attenuation even in anechoic setting. I am not convinced that this works at all as proposed in realistic settings. 2. The experimental section also is very simplistic, they have simple signal model – based on that they generate data and show results. The results are naive as the simulations are all in 2D and it makes a difference going from 2D to 3D. 3. The real data part is also not how it should technically be done. Most importantly localization results are shown on 2 speakers which not all that convincing. 4. The authors assume that there is no leakage of a source to surrounding locations, i.e. point source. This can only happen in simulations and seldom works in practice. 5. Most importantly reverberation has not been considered. Due to reverberation phantom source are created which limits the utility of such an approach as it becomes difficult to isolate a phantom source from a real source due to reflections. 6. Reporting localization and separation results only on simulated data was fine perhaps a decade ago. Reporting on real data is essential for testing the validity of all the assumptions made. In this particular case the reported experiments are rather simplistic and not convincing that this approach can be applied in practice.

Correctness: Claims are correct with the strong assumptions which render the proposed approach to be limited in practice.

Clarity: Yes

Relation to Prior Work: Yes

Reproducibility: Yes

Additional Feedback:

[Author Response · NeurIPS 2020]

We would like to thank the reviewers for their thoughtful comments.

*General comments*:

**Train/test data split, generalization to real results (R3, R4)**: Our test and train sets are completely independent and there are no overlapping identities between train/test splits—our network is *only* trained on VCTK data but performs well on real speakers in unseen environments, as demonstrated in the supplementary video. As requested, we ran numerical results on a test set of real recorded data, and found a median SI-SDRi of $10.7\,\mathrm{dB}$ on mixtures of 3 speakers, higher than Oracle IBM $10.2\,\mathrm{dB}$ on the same data. Full comparisons and results will be reported in the final version.

**Elevation Angle (R1, R4)**: Many real-world situations, like the ones in the supplementary video, can be well approximated by an azimuthal-only model. Although our work only focuses on localization and separation by azimuthal angle, we will add the note by R1 about our model's assumption regarding the relationship between elevation angle and TDOA. We will additionally add a discussion on possible extension of the proposed model to handle elevation.

**Experimental data (R1, R3)**: Full details on rendering parameters will be added in the final-version.

*Reviewer #1 (R1)*:

**Eq. 6**: $\mathbf{s}'_i$ is computed by using Eq. 3 the same way that $\mathbf{x}'$ is computed, using the TDOA.

**Conv-TasNet**: We modified the single-channel Conv-TasNet to handle multi channels by changing the number of input channels and output channels on the first and last layer of the network.

**Is the method expected to generalize to any array geometry?**: We need to use a different model trained on each different array geometry. **Post-processing**: All synthetic results and numbers are reported without post-processing. We will add qualitative real results to show the comparison.

*Reviewer #2 (R2)*:

**Beamforming**: We thank the reviewer for pointing out the analogy to tunable-width beamforming, which we will add in the related work. The benefit of our method over tunable-width beamformers is an ability to select a certain type of audio (e.g., speech), which is otherwise impossible to separate if the speaker is spatially close to a noise source. We will also add a comparison to state-of-the art beamforming in the experiments section as suggested.

**Global Sweep and Tracking**: Although we use binary search as a global sweep, our method can also be used as suggested to perform local sweeps in subsequent time steps. We will add a discussion of this.

**Usefulness and Speed**: It is true that the network is not specifically designed for real time processing, but it can run on chunks as small as $0.5\,\mathrm{s}$, making it practical for many use cases such as smart home devices, media production, or meeting transcriptions. Real time processing is an exciting future work, which could use slightly different network architectures while maintaining our core idea of a target angle and variable window size.

*Reviewer #3 (R3)*: Please confirm this review is for our paper since the summary is for a lip sync paper.

**Different sampling rate**: When running at $16\,\mathrm{kHz}$, we found that the median angular error was worse by $1.3°$, while the separation results were worse by $2.23\,\mathrm{dB}$. We can add these lower sample rate experiments and comparisons to SOTA in the paper. We believe that the ability to operate at higher sample rates is a major benefit of our method that will allow extensions to other source types, for instance, high-resolution music signals.

**Clean target audio**: When we say "clean sample," we just mean we need each source separately since our method is supervised. **Why cannot the samples be mixed with reverberations?** The target audio is mixed with reverberations, which is then used as the ground truth target. More clarifications will be included in the final-version.

**Oracle IRM**: Yes, the low result is correct. We used the evaluation code provided as a part of SiSEC 2018 campaign, and confirmed qualitatively that IRM is poor. Because the mixtures contain high levels of background noise, the IRM contains mostly phase information from the background. **Is post-processing used in synthetic testing?** No, it is not.

*Reviewer #4 (R4)*:

**Unrealistic simple signal model and experiments**: We would like to correct some misunderstandings regarding our rendering method. Signal attenuation and reverb are both considered and modeled in the synthetic data with the `pyroomacoustics` library. Furthermore, both real and synthetic results show that phantom sources are not picked up.

**Leakage of Source**: The variable window size is motivated by the non-point source nature of sources. Although the synthetic data is modeled as point sources, the real results show that we can handle moving and non-point sources. We also show localization and separation on 4 real sources, not just 2.

**Localization and separation results only on simulated data**: We note that many recent papers on spatial audio such as [1] from 2020 and [2] from 2018 show results only on synthetically rendered data.

**"Not convincing that this approach can be applied in practice"**: The real results we have shown are not cherry picked, and are natural scenarios with speakers and environments not seen during training. This represents evidence that it can indeed work in practice. In addition, our real results are actually shown on up to 4 real speakers, not only 2.

[1] Luo, Yi, et al. "End-to-end microphone permutation and number invariant multi-channel speech separation." *ICASSP*. 2020.
[2] Johnson, Daniel, et al. "Latent Gaussian activity propagation: using smoothness and structure to separate and localize sounds in large noisy environments." *NeurIPS*. 2018.


[Meta-Review · NeurIPS 2020]

This paper designs a network which can map the multi-channel audio from an array to the audio of a source at desired angular region. There has been active discussions after the author rebuttal. In the end, all reviewers agree that there is sufficient novelty in the methodology. There have also been minor concerns on the experimental setup being less realistic, not demonstrating whether the method is robust under more challenging scenarios. But reviewers also reached the consensus that the current result is significant enough for a conference paper. We hope the authors take advantage of the insightful reviews in future work.